# Treadmill Exercise Reduces Neuroinflammation, Glial Cell Activation and Improves Synaptic Transmission in the Prefrontal Cortex in 3 × Tg-AD Mice

**DOI:** 10.3390/ijms232012655

**Published:** 2022-10-21

**Authors:** Lianwei Mu, Dongdong Xia, Jiajia Cai, Boya Gu, Xiaojie Liu, Vladislav Friedman, Qing-Song Liu, Li Zhao

**Affiliations:** 1Department of Exercise Physiology, Guangzhou Sport University, Guangzhou 510500, China; 2Key Laboratory of Physical Fitness and Exercise, Ministry of Education, Beijing Sport University, Beijing 100084, China; 3Department of Pharmacology and Toxicology, Medical College of Wisconsin, 8701 Watertown Plank Road, Milwaukee, WI 53226, USA

**Keywords:** exercise, Aβ oligomer, GSK3β, inflammatory factor, glial cell, synaptic plasticity, 3 × Tg-AD mice

## Abstract

Physical exercise improves memory and cognition in physiological aging and Alzheimer’s disease (AD), but the mechanisms remain poorly understood. Here, we test the hypothesis that Aβ oligomer accumulation, neuroinflammation, and glial cell activation may lead to disruption of synaptic transmission in the prefrontal cortex of 3 × Tg-AD Mice, resulting in impairment of learning and memory. On the other hand, treadmill exercise could prevent the pathogenesis and exert neuroprotective effects. Here, we used immunohistochemistry, western blotting, enzyme-linked immunosorbent assay, and slice electrophysiology to analyze the levels of GSK3β, Aβ oligomers (Aβ dimers and trimers), pro-inflammatory cytokines (IL-1β, IL-6, and TNFα), the phosphorylation of CRMP2 at Thr514, and synaptic currents in pyramidal neurons in the prefrontal cortex. We show that 12-week treadmill exercise beginning in three-month-old mice led to the inhibition of GSK3β kinase activity, decreases in the levels of Aβ oligomers, pro-inflammatory cytokines (IL-1β, IL-6, and TNFα), and the phosphorylation of CRMP2 at Thr514, reduction of microglial and astrocyte activation, and improvement of excitatory and inhibitory synaptic transmission of pyramidal neurons in the prefrontal cortex of 3 × Tg-AD Mice. Thus, treadmill exercise reduces neuroinflammation, glial cell activation and improves synaptic transmission in the prefrontal cortex in 3 × Tg-AD mice, possibly related to the inhibition of GSK3β kinase activity.

## 1. Introduction

A prominent feature of Alzheimer’s disease (AD) is the loss of synapses [1]. Synapse loss in AD is expected to cause the disruption of synaptic transmission, represented by the change of miniature excitatory and inhibitory postsynaptic currents (mEPSCs and mIPSCs). Previous studies show that synaptosomal-associated protein 25 (*SNAP-25*) and synapsin 2 (*SYN2*) genes associated with synaptic vesicle release, were among the most prominently downregulated differentially expressed genes in the prefrontal cortex of humans with Alzheimer’s disease (AD) [2,3]. While glial fibrillary acidic protein (*GFAP*) and Crystallin Alpha B (*CRYAB*), both of which are involved in gliosis and glial-related pathology associated with neurodegeneration, were differentially upregulated [2,3]. In the postmortem brain of AD patients, abundant inflammatory cytokines, reactive astrocytes and microglial cells surrounding amyloid-β (Aβ) plaques can readily be seen [4,5]. Additionally, it has been shown that microglia and astrocytes were activated in the prefrontal cortex and hippocampus during the early stage of the disease and significantly increased with the disease progresses in different transgenic mouse models of AD, including Tg2576 [6], APP/PS1 [7], 5 × FAD [8], and 3 × Tg-AD [9,10] mice. Lastly, Aβ oligomers mediate early inflammatory response by activating astrocytes or microglia [11,12]. On the one hand, the activated microglia are able to internalize and degrade Aβ and reduce Aβ mediated neurotoxicity in vivo and in vitro; the astrocytes and microglia activated by Aβ release proinflammatory factors, including interleukin-1β (IL-1β), interleukin-6 (IL-6), and tumor necrosis factor alpha (TNF-α) that lead to neuroinflammation and neuron death [7,13,14,15,16]. Chronic treatments of AD mice with an IL-1 receptor (IL-1R) or TNF-α blocking antibodies significantly reduce the activity of Glycogen synthase kinase-3β (GSK3β), brain inflammatory responses, cognitive deficits, tau pathology, and both fibrillar and oligomeric forms of Aβ [17,18,19]. GSK3β is a major regulator of the balance between proinflammatory and anti-inflammatory signaling in the brain [20]. GSK3 agonists promote the production of inflammatory factors and cell migration, while GSK3 inhibitors attenuate the production of inflammatory molecules by activated glial cells (microglia, astrocytes) and provide strong protection from inflammation-induced neurotoxicity [21,22]. Furthermore, collapsing response mediator protein 2 (CRMP2), a microtubule-binding protein, serves important functions in axon growth, synaptic plasticity, vesicular trafficking, and learning [23,24]. Phosphorylation of CRMP2 by GSK3 inhibits CRMP2 activity [25], and levels of the phosphorylated CRMP2 were increased in the brains of AD patients and AD mouse models [26,27]. These results indicate that GSK-3β-mediated neuroinflammation, glial cell activation, and impaired synaptic transmission may be involved in the pathogenesis of AD.

We and others have shown that physical exercise decreases Aβ burden and alleviates synapse loss and impairment of learning and memory in mouse models [28,29,30]. Meanwhile, exercise reduces the activation of microglia and astrocytes and neuroinflammation and preserves cognitive function in 3 × Tg-AD mice [31]. GSK3 modulates a wide range of cellular metabolic processes, including insulin sensitivity, glucose metabolism, and cell survival [32]. Inhibitors of GSK3β provide therapeutic benefit in models of AD [33], Parkinson’s disease (PD) [34], and spinal cord lesion [22]. In the present study, we examine whether 12 weeks of treadmill exercise affects neuroinflammation, glial cell activation and synaptic transmission of the prefrontal cortex in 3 × Tg-AD Mice. We found that 12 weeks of treadmill exercise decreased the levels of Aβ oligomers (Aβ trimers), inhibited the kinase activity of GSK3β, down-regulated immune response, and improved synaptic transmission in the prefrontal cortex of 3 × Tg-AD Mice.

## 2. Results

### 2.1. Effects of Treadmill Exercise on Aβ Plaques and Aβ Oligomers (Aβ Dimers and Aβ Trimers) Levels of the Prefrontal Cortex in 3 × Tg-AD Mice

To test the role that treadmill exercise has on Aβ plaques and Aβ oligomers (Aβ dimers and Aβ trimers), non-Tg control mice and 3 × Tg-AD mice received 12 weeks of treadmill exercise or non-exercise control treatment beginning at three months of age (2 × 2 factorial design: genotype vs. exercise). After the 12-week training, we measured the levels of Aβ plaques and Aβ oligomers (Aβ dimers and Aβ trimers) in the prefrontal cortex by immunohistochemical analysis and western blotting (Figure 1A). We show that Aβ plaques were not detected in the prefrontal cortex in six-month-old 3 × Tg-AD mice (Figure 1B). Aβ oligomers (Aβ dimers and Aβ trimers) play an important role in AD pathophysiology [35,36,37]. Two-way ANOVA indicated that genotype and treadmill exercise had significant effects on the expression of Aβ trimers (genotype: F_1,28_ = 77.7, *p* < 0.001; treadmill exercise: F_1,28_ = 29.3, *p* < 0.001; genotype × treadmill exercise interaction: F_1,28_ = 20.3, *p* < 0.001; Figure 1C,D) in the prefrontal cortex. Tukey’s post hoc tests indicated that the expression of Aβ trimers were significantly increased in the 3 × Tg-AD control group compared to the non-Tg control group (*p* < 0.001; Figure 1D). Treadmill exercise prevented the increase in the expression of Aβ trimers of the prefrontal cortex in 3 × Tg-AD mice (*p* < 0.001; Figure 1D). However, we did not detect low molecular weight Aβ oligomer (Aβ dimers, Figure 1C). These results suggest that treadmill exercise ameliorates the level of Aβ oligomers (Aβ trimers) in the prefrontal cortex in six-month-old 3 × Tg-AD mice.

### 2.2. Treadmill Exercise Inhibited the Kinase Activity of GSK3β of the Prefrontal Cortex in 3 × Tg-AD Mice

Aβ can aberrantly activate GSK3β [38]. To determine the effect of treadmill exercise on GSK3β activity, we assessed the phosphorylation of GSK3β at Tyr216 and Ser9 (Figure 2A,B). GSK3β is thought to be constitutively activated by autophosphorylation at Tyr216 and inactivated by phosphorylation at Ser9 [39]. Two-way ANOVA revealed that genotype and treadmill exercise had significant effects on the phosphorylation of GSK3β at Tyr216 (genotype: F_1,28_ = 9.1, *p* = 0.005; treadmill exercise: F_1,28_ = 25.0, *p* < 0.001; genotype × treadmill exercise interaction: F_1,28_ = 6.9, *p* = 0.014; Figure 2C) and Ser9 (genotype: F_1,28_ = 9.1, *p* = 0.005; treadmill exercise: F_1,28_ = 15.5, *p* < 0.001; genotype × treadmill exercise interaction: F_1,28_ = 5.2, *p* = 0.030; Figure 2D) of the prefrontal cortex. Tukey’s post hoc tests indicated that phosphorylation of GSK3β at Tyr216 was significantly up-regulated in the 3 × Tg-AD control group compared to that of the non-Tg control group (*p* < 0.001; Figure 2C). In contrast, the phosphorylation of GSK3β at Ser9 in the prefrontal cortex was significantly down-regulated in the 3 × Tg-AD control group compared to that of the non-Tg control group (*p* < 0.001; Figure 2D). Treadmill exercise prevented the increased phosphorylation of GSK3β at Tyr216 (*p* < 0.001; Figure 2C) and the decreased phosphorylation of GSK3β at Ser9 (*p* < 0.001; Figure 2D) in 3 × Tg-AD mice. The results demonstrated that treadmill exercise inhibited the kinase activity of GSK3β.

### 2.3. Treadmill Exercise Reduced the Activation of Microglia and Astrocytes of the Prefrontal Cortex in 3 × Tg-AD Mice

GFAP and CD68 were used as markers for reactive astrocytes and microglia, respectively [40]. To determine the effect of treadmill exercise on the activation of astrocytes and microglia, we assessed the expression of GFAP and CD68 of the prefrontal cortex by Western Blot (Figure 3A). Two-way ANOVA revealed that genotype and treadmill exercise had significant effects on the expression of GFAP (genotype: F_1,28_ = 4.9, *p* = 0.035; treadmill exercise: F_1,28_ = 14.7, *p* < 0.001; genotype × treadmill exercise interaction: F_1,28_ = 13.2, *p* = 0.001; Figure 3B) and CD68 (genotype: F_1,28_ = 3.3, *p* = 0.08; treadmill exercise: F_1,28_ = 8.6, *p* = 0.007; genotype × treadmill exercise interaction: F_1,28_ = 8.4, *p* = 0.007; Figure 3C) in the prefrontal cortex. Tukey’s post hoc tests indicated that the levels of GFAP (*p* < 0.001; Figure 3B) and CD68 (*p* = 0.003; Figure 3C) in the prefrontal cortex were significantly increased in the 3 × Tg-AD control group compared to the non-Tg control group. Treadmill exercise prevented the increase in GFAP levels (*p* < 0.001; Figure 3B) and CD68 (*p* < 0.001; Figure 3C) in the prefrontal cortex in 3 × Tg-AD mice.

We also performed immunohistochemical analysis for GFAP to examine the number of GFAP positive glial cells in the prefrontal cortex (Figure 3D). Two-way ANOVA indicated that genotype and treadmill exercise had significant main effects on the number of GFAP positive glial cells (genotype: F_1,36_ = 21.3, *p* < 0.001; treadmill exercise: F_1,36_ = 14.1, *p* < 0.001; Figure 3E), but there was not a significant interaction between genotype and treadmill exercise on the number of GFAP positive glial cells (genotype × treadmill exercise interaction: F_1,36_ = 1.6, *p* = 0.219; Figure 3E). Tukey’s post hoc tests indicated that the number of GFAP positive glial cells in the prefrontal cortex were significantly increased in the 3 × Tg-AD control group compared to that of the non-Tg control group (*p* < 0.001; Figure 3E). Treadmill exercise prevented the increase in the number of GFAP positive glial cells in the prefrontal cortex in 3 × Tg-AD mice (*p* = 0.001; Figure 3E). Together, these results suggest that there was significantly more extensive activation of astrocytes and microglia in the prefrontal cortex of six-month-old 3 × Tg-AD mice compared to control mice, and that treadmill exercise prevented the activation of astrocytes and microglia.

### 2.4. Treadmill Exercise Decreased the Concentration of Pro-Inflammatory Cytokines (IL-1β, IL-6, and TNFα) of the Prefrontal Cortex in 3 × Tg-AD Mice

Activated microglia and astrocytes secrete increased levels of pro-inflammatory cytokines, such as IL-1β, IL-6, and TNFα [41]. To quantify the release of pro-inflammatory cytokines, AD-relevant cytokines IL-1β, IL-6, and TNFα were measured using ELISA. Two-way ANOVA revealed that genotype and treadmill exercise had significant effects on the concentration of IL-1β (genotype: F_1,28_ = 50.4, *p* < 0.001; treadmill exercise: F_1,28_ = 24.0, *p* < 0.001; genotype × treadmill exercise interaction: F_1,28_ = 19.1, *p* < 0.001; Figure 4A), IL-6 (genotype: F_1,28_ = 19.6, *p* < 0.001; treadmill exercise: F_1,28_ = 11.7, *p* = 0.002; genotype × treadmill exercise interaction: F_1,28_ = 5.3, *p* = 0.029; Figure 4B), and TNFα (genotype: F_1,28_ = 55.6, *p* < 0.001; treadmill exercise: F_1,28_ = 29.2, *p* < 0.001; genotype × treadmill exercise interaction: F_1,28_ = 18.6, *p* < 0.001; Figure 4C) in the prefrontal cortex. Tukey’s post hoc tests indicated that the concentration of IL-1β (*p* < 0.001; Figure 4A), IL-6 (*p* < 0.001; Figure 4B), and TNFα (*p* < 0.001; Figure 4C) in the prefrontal cortex were significantly increased in the 3 × Tg-AD control group compared to those of the non-Tg control group. Treadmill exercise pretreatment blocked the increase in the concentration of IL-1 β (*p* < 0.001; Figure 4A), IL-6 (*p* < 0.001; Figure 4B), and TNFα (*p* < 0.001; Figure 4C) in the prefrontal cortex in 3 × Tg-AD mice. Thus, treadmill exercise reduced the concentrations of pro-inflammatory cytokines (IL-1β, IL-6, and TNFα), which are likely mediated by decreasing the activation of astrocytes and microglia.

### 2.5. Treadmill Exercise Decreased the Levels of Phosphorylation of CRMP2 at Thr514 of the Prefrontal Cortex in 3 × Tg-AD Mice

CRMP2 is a microtubule-associated protein that is hyperphosphorylated by GSK3β in AD [25,26]. As treadmill exercise inhibited the kinase activity of GSK3β, we also examined the phosphorylation level of CRMP2 at Thr514 in the prefrontal cortex in 3 × Tg-AD mice (Figure 4D). Two-way ANOVA revealed that genotype and treadmill exercise had significant effects on the phosphorylation of CRMP2 at Thr514 (genotype: F_1,28_ = 27.5, *p* < 0.001; treadmill exercise: F_1,28_ = 18.9, *p* < 0.001; genotype × treadmill exercise interaction: F_1,28_ = 6.0, *p* = 0.021; Figure 4E). Tukey’s post hoc tests indicated that the phosphorylation of CRMP2 at Thr514 in the prefrontal cortex was significantly up-regulated in the 3 × Tg-AD control group compared to the non-Tg control group (*p* < 0.001; Figure 4E). Treadmill exercise prevented the up-regulation of the CRMP2 phosphorylation at Thr514 in the prefrontal cortex in 3 × Tg-AD mice (*p* < 0.001; Figure 4E).

### 2.6. Treadmill Exercise Ameliorated Synaptic Transmissi on Dysfunction of the Prefrontal Cortex in 3 × Tg-AD Mice

To investigate the effects of treadmill exercise on synaptic transmission in the prefrontal cortex in 3 × Tg-AD mice, we recorded miniature excitatory postsynaptic currents (mEPSCs, Figure 5A–C) and miniature inhibitory postsynaptic currents (mIPSCs, Figure 5D–F) from pyramidal neurons in ex vivo prefrontal cortex slices. Two-way ANOVA revealed that genotype and treadmill exercise had significant main effects on the mean frequency of mEPSCs (genotype: F_1,56_ = 11.7, *p* < 0.001; treadmill exercise: F_1,56_ = 11.1, *p* = 0.002; Figure 5B) and mIPSCs (genotype: F_1,56_ = 23.1, *p* < 0.001; treadmill exercise: F_1,56_ = 22.2, *p* < 0.001; Figure 5E), and the mean amplitude of mEPSCs (genotype: F_1,56_ = 33.5, *p* < 0.001; treadmill exercise: F_1,56_ = 20.8, *p* < 0.001; Figure 5C) and mIPSCs (genotype: F_1,56_ = 80.0, *p* < 0.001; treadmill exercise: F_1,56_ = 21.3, *p* < 0.001; Figure 5F), but there was no significant interaction between genotype and treadmill exercise on the mean frequency of mEPSCs (genotype × treadmill exercise interaction: F_1,56_ = 0.07, *p* = 0.795) and mIPSCs (genotype × treadmill exercise interaction: F_1,56_ = 1.6, *p* = 0.215), and the mean amplitude of mEPSCs (genotype × treadmill exercise interaction: F_1,56_ = 0.8, *p* = 0.368) and mIPSCs (genotype × treadmill exercise interaction: F_1,56_ = 0.4, *p* = 0.519). Tukey’s post hoc tests indicated the mean frequency of mEPSCs (*p* = 0.014; Figure 5B), the mean amplitude of mEPSCs (*p* < 0.001; Figure 5C), the mean frequency of mIPSCs (*p* < 0.001; Figure 5E), and the mean amplitude of mIPSCs (*p* < 0.001; Figure 5F) in the prefrontal cortex were significantly reduced in the 3 × Tg-AD control group compared to that of the non-Tg control group. Treadmill exercise prevented the reduction of the mean frequency of mEPSCs (*p* = 0.012; Figure 5B), the mean amplitude of mEPSCs (*p* < 0.001; Figure 5C), the mean frequency of mIPSCs (*p* < 0.001; Figure 5E), and the mean amplitude of mIPSCs (*p* < 0.001; Figure 5F) of the prefrontal cortex in 3 × Tg-AD mice. Meanwhile, treadmill exercise increased the mean frequency of mEPSCs (*p* = 0.034; Figure 5B), the mean amplitude of mEPSCs (*p* = 0.018; Figure 5C), the mean frequency of mIPSCs (*p* = 0.012; Figure 5E), and the mean amplitude of mIPSCs (*p* = 0.007; Figure 5F) in the prefrontal cortex in non-Tg mice.

### 2.7. GSK3β Inhibitor (ARA-014418) Restored the Synaptic Transmission Dysfunction of the Prefrontal Cortex in 3 × Tg-AD Mice

We examined whether microinjections of GSK3β inhibitor (ARA-014418) in the left lateral ventricle affected the synaptic transmission of the prefrontal cortex in 3 × Tg-AD mice. Mice were placed in a stereotaxic device and allowed to recover for 5–7 days prior to the start of drug infusion. Then, the mice were microinjected into the left lateral ventricle with 2 μL of either artificial cerebrospinal fluid (Vehicle, control group) or GSK3β solution (1.2 μM). Drug treatment was administered by twice-daily injections, 6 h apart for 3 successive days. The next day, mice were used for electrophysiology experiments (Figure 6A). We found that lateral ventricle injection of GSK3β inhibitor caused significant increases in the mean frequency (t_1,28_ = 3.168, *p* = 0.004; Figure 6B) and amplitude (t_1,28_ = 4.8887, *p* < 0.001; Figure 6C) of mEPSCs in 3 × Tg-AD mice. Meanwhile, ventricle injection of GSK3β inhibitor caused significant increases in the mean frequency (t_1,28_ = 3.925, *p* < 0.001; Figure 6D) and amplitude (t_1,28_ = 3.568, p = 0.001; Figure 6E) of mIPSCs in 3 × Tg-AD mice. This result indicated that inhibition of GSK3β kinase activity promotes neuroprotection by enhancing GABAergic inhibition and glutamatergic excitation of pyramidal neurons in the prefrontal cortex in 3 × Tg-AD Mice.

## 3. Discussion

Here, we have demonstrated that Aβ oligomers (Aβ trimers) levels were found to be significantly increased in the prefrontal cortex in six-month-old 3 × Tg-AD mice, but we did not detect extracellular amyloid plaques or Aβ dimers oligomers at this age (Figure 1C,D). Treadmill exercise led to decreases in the levels of soluble Aβ trimers, kinase activity of GSK3β, the concentrations of pro-inflammatory cytokines (IL-1β, IL-6, and TNFα), and the phosphorylation of CRMP2 at Thr514 in the prefrontal cortex of 3 × Tg-AD mice. Additionally, treadmill exercise prevented the activation of microglia and astrocytes and decreases in GABAergic inhibition and glutamatergic excitation in the prefrontal cortex in 3 × Tg-AD mice. Finally, lateral ventricle injection of a GSK3β inhibitor enhanced GABAergic inhibition and glutamatergic excitation of pyramidal neurons in the prefrontal cortex in 3 × Tg-AD Mice. Taken together, these results suggest that treadmill exercise reduces the accumulation of soluble Aβ trimers, neuroinflammation, microglial and astrocyte activation and restores synaptic transmission in the prefrontal cortex in 3 × Tg-AD mice.

We did not detect Aβ plaques in the prefrontal cortex of the six-month-old 3 × Tg-AD mice, which may be in the early stage of the disease. Aβ plaques and Aβ oligomers activate generic cytotoxicity pathways, but soluble species of Aβ might play a larger neurotoxic role in AD pathophysiology than Aβ plaques in the early stage of the AD. Aβ dimers, Aβ trimers, and spherical oligomers are the major forms of Aβ oligomers [35,42]. Aβ trimers are generally considered to have conspicuous toxicity to neurons, but the toxicity of Aβ dimers was very weak [42,43]. In the present study, Aβ trimers, but not Aβ dimers, were significantly increased in in the prefrontal cortex in six-month-old 3 × Tg-AD mice. It is likely that Aβ trimers play a critical role in early pathogenesis of 3 × Tg-AD mice. Meanwhile, Aβ dimers may produce in the late stage of AD and influence the late-stage of AD pathogenesis. Previous studies have shown that Aβ dimers can be found in soluble protein extracts in Tg2576 and J20 AD mice aged 10–14 months [44,45]. Treadmill exercise also decreased the levels of Aβ trimers in the prefrontal cortex in 3 × Tg-AD mice, suggesting that exercise can effectively ameliorate pathology in the early stage of the AD.

GSK3β, a multifunctional serine/threonine protein kinase [46], is critically involved in the molecular pathology of many neurodegenerative diseases. Oligomeric and fibrillar Aβ peptides significantly decreased the level of GSK3β phosphorylated at Ser9, which suggests that both Aβ peptides induce GSK3β activation [47,48]. GSK3β is thought to be constitutively activated by autophosphorylation at Tyr216 and inactivated by phosphorylation at Ser9 [49]. Here, we have shown that treadmill exercise decreased levels of phosphorylation of GSK3β at Tyr216, and increased levels of phosphorylation of GSK3β at Ser9, and inhibited the kinase activity of GSK3β (Figure 2C,D). GSK3β plays a key role in regulating communication between neurons and glial cells by altering the activation of glial cells and synaptic transmission [21,50]. GFAP and CD68 were used as markers of astrocytic and microglial activation, respectively [51,52]. We found that the number of GFAP positive glial cells, and the levels of GFAP and CD68 in the prefrontal cortex were significantly increased in the 3 × Tg-AD mice. Treadmill exercise decreased the number of GFAP positive glial cells, and the levels of GFAP and CD68 in the prefrontal cortex in the 3 × Tg-AD mice. These results indicate that treadmill exercise inhibited the hyperactivation of astrocytic and microglial responses by dampening the activity of GSK3β. Both activated astrocytes and microglia can release inflammatory factors that lead to the neuroinflammation [53]. Therefore, we further examined the levels of pro-inflammatory cytokines IL-1β, IL-6, and TNFα, and found that the concentrations of IL-1β, IL-6, and TNFα was significantly increased in the prefrontal cortex of 3 × Tg-AD mice, but this increase was blocked by treadmill exercise. It is likely that treadmill exercise suppresses glia-mediated inflammation by down-regulating activity of GSK3β in the prefrontal cortex of 3 × Tg-AD mice.

IL-1β, IL-6, and TNFα binding to their receptor complexes can lead to activation of GSK3β, which can regulate the phosphorylation of the CRMP2 to participate in synaptic plasticity [54,55,56]. CRMP2 is a microtubule-binding protein that serves important functions in synaptic plasticity, vesicular trafficking, and learning [23,24]. However, CRMP2 phosphorylation at Thr514 decreases its tubulin-binding activity, potentially leading to axonal degeneration [57]. Indeed, we found that the phosphorylation of CRMP2 at Thr514 in the prefrontal cortex was significantly increased in 3 × Tg-AD mice. Treadmill exercise prevented the up-regulation of the phosphorylation of CRMP2 at Thr514 in the prefrontal cortex in 3 × Tg-AD mice. Our previous study showed that treadmill exercise pretreatment strengthened structural synaptic plasticity by increasing the number of synapses, synaptic structural parameters, expression of synaptophysin (Syn, a presynaptic marker), axon length, dendritic complexity, and the number of dendritic spines [28]. It is thus likely that treadmill exercise reversed structural synaptic plasticity by inhibiting the phosphorylation of CRMP2 at Thr514 in the prefrontal cortex in 3 × Tg-AD mice. To better reflect changes in synaptic transmission efficiency, mEPSCs and mIPSCs were recorded from pyramidal neurons of the prefrontal cortex. We found that the GABAergic inhibition and glutamatergic excitation were significantly reduced in pyramidal neurons of the prefrontal in 3 × Tg-AD mice, while treadmill exercise prevented the decrease in the GABAergic inhibition and glutamatergic excitation. Additionally, treadmill exercise improved GABAergic inhibition and glutamatergic excitation in pyramidal neurons of the prefrontal cortex in non-Tg mice, indicating that treadmill exercise can enhance synaptic plasticity in non-Tg mice. We extended these findings by showing that lateral ventricle injection of GSK3β inhibitor strengthened GABAergic inhibition and glutamatergic excitation in pyramidal neurons of the prefrontal cortex in 3 × Tg-AD mice (Figure 6). These results suggest that treadmill exercise improved synaptic transmission by down-regulating the activity of GSK3β in the prefrontal cortex of 3 × Tg-AD mice.

## 4. Materials and Methods

### 4.1. Animals

Male Triple-transgenic Alzheimer’s disease (3 × Tg-AD) and C57BL/6J were used in the present study. 3 × Tg-AD mice (harboring PS1M146V knockin, APPswe, and TauP301L transgenes) were acquired from The Jackson Laboratory. Age and sex matched C57BL/6J mice were employed as non-transgenic (non-Tg) controls. The study was performed on 3-month-old 3 × Tg-AD (n = 72) and non-Tg mice (n = 60). All animals were given ad libitum access to food and water in a room maintained at controlled temperature (23 ± 1 °C) and humidity (40–60%), with a 12 h light-dark cycle. All animal maintenance and use were in accordance with protocols approved by the ethical committee of Beijing Sport University (2015015).

### 4.2. Chemical Reagents

Picrotoxin, Tetrodotoxin citrate (TTX) and all other common chemicals were obtained from Sigma-Aldrich (St. Louis, MO, USA). 6-cyano-7-nitroquinoxaline-2,3-dione disodium salt (CNQX), D-(−)-2-Amino-5-phosphonopentanoic acid (D-AP5), and GSK3β inhibitor (ARA-014418) were obtained from Tocris Bioscience (Ellisville, MO, USA). Picrotoxin was ultrasonic to dissolve in ACSF before use. TTX, CNQX and D-AP5 were dissolved in ddH2O as stock solutions and stored at −20 °C freezer. ARA-014418 was dissolved in DMSO at stock concentrations of 50 mM and stored at −80 °C freezer.

### 4.3. Treadmill Exercise Protocol

At 3 months of age, mice were randomly assigned into four groups (n = 30/group): non-Tg control group, non-Tg exercise group, 3 × Tg-AD control group, 3 × Tg-AD exercise group. All of the exercise groups were allowed to adapt to treadmill running for 30 min on 3 consecutive days (first day at 5 m/min; second and third day at 10 m/min). Then, the mice were subjected to a treadmill exercise protocol at the speed of 12 m/min on a 0° slope for 10 min, then 15 m/min for 50 min. In total, mice were trained for 1 h per day, 5 days per week, for a total of 12 weeks. According to our test by an enclosed single-lane treadmill attached to an Oxymax/Comprehensive Laboratory Animal Monitoring System, this long-duration training protocol maintains exercise intensity from 50% to 65% of VO2 max. Mice in the non-Tg control and 3 × Tg-AD control groups were left on the treadmill, without running, for the same period as exercise groups. Then, we examined the histological abnormalities by Immunohistochemistry analysis, Western Blot, Enzyme-linked immunosorbent assay (ELISA), Slice electrophysiology (Figure 1A).

### 4.4. Animal Surgery and Microinjection Procedures

At 6 months of age, 3 × Tg-AD mice were anesthetized with ketamine (90 mg/kg, i.p.) and xylazine (10 mg/kg, i.p.) and placed in a stereotaxic device (David Kopf Instruments, Tujunga, CA, USA). 22-gauge guide cannula was implanted to the left lateral ventricle (from bregma, anteroposterior, −0.3 mm; mediolateral, −1.0 mm; dorsoventral, −2.0 mm) [58,59,60]. Mice were allowed to recover for 5–7 days prior to the start of drug infusion. Drug infusions were done by a 27-gauge needle (0.2 mm longer than the guide cannula) attached to a Hamilton micro-syringe via polyethylene tube. Mice were microinjected into the left lateral ventricle with 2 μL of either artificial cerebrospinal fluid (Vehicle, Vehicle group, n = 6) or solution of GSK3β inhibitor (1.2 mΜ; ARA-014418 group, n = 6). Infusions lasted for 60 s and the cannula was left in place for an additional 5 min to avoid the backflow of the solution. Drug was administered by twice-daily injections, 6 h apart for 3 successive days [61]. The next day, mice were used for electrophysiology experiments.

### 4.5. Immunohistochemistry Analysis

Mice were deeply anesthetized with isoflurane and perfused through the left ventricle of the heart with 4% paraformaldehyde in 0.1 M sodium phosphate buffer (PBS), pH 7.4. The brains were removed and postfixed by immersion in the same fixative for 24 h at 4 °C. They were then cryoprotected in increasing concentrations of sucrose (20% and 30%) in 0.1 M PBS at 4 °C, frozen in liquid nitrogen, and stored at −80 °C until use. Coronal sections of frozen brains were cut at 40 um thickness with a cryostat (Leica, Wetzlar, Germany). Sections were then washed with PBS for three times, 10 min each. Sections were washed 10 min after formic acid repair (70% formic acid, 10 min), inactivated endogenous peroxidase (3% H_2_O_2_ of 100% methanol solution) and antigen repair (20 ug/mL proteinase K solution, 37 °C, 20 min) (Proteinase K, Sigma, USA). Sections were blocked at RT with 5% BSA for 30 min. Sections were then incubated at 4 °C for 2 d in the primary antibodies diluted at the appropriate concentrations in the same solutions used for blocking (Aβ (6E10: 803004), 1:1000, Convance, Burlington, VT, USA; GFAP, 1:100, Abcam: ab7260, Boston, MA, USA). Then, sections were incubated in the appropriate secondary antibodies: goat anti-mouse IgG-horseradish peroxidase (1:500, Proteintech Group, Rosemont, IL, USA) or goat anti-rabbit IgG-horseradish peroxidase (1:500, Proteintech Group, USA). Sections were then incubated in the peroxidase-conjugated streptavidin (Vector laboratories, Newark, CA, USA). Immunoreactivity was detected with a DAB kit (SK-4105, Vector laboratories, USA) and the reaction was stopped with water. They were dehydrated in an ascending ethanol series (70%, 80%, 95%, 95%, 100% and 100%), cleared in xylene two times, and coverslips were applied.

### 4.6. Western Blot

Mice were anesthetized by isoflurane inhalation and decapitated. Protein extraction and western blot analysis of the prefrontal cortex were performed according to our recent study [28]. Protein levels were detected with primary antibody (GAPDH, 1:10,000, Abcam: ab8245, Boston, MA, USA; Aβ oligomers (4G8); 1:800, Biolegend:800708, Santiago, AR, USA; p-GSK3β-Tyr216, 1:3000, Abcam: ab68476; p-GSK3β-Ser9, 1:5000, Abcam: ab131097; GSK3β, 1:5000, Abcam: ab93926; CD68, 1:1000, Abcam: ab125212; GFAP, 1:5000, Abcam: ab7260; p-CRMP2-Thr514, 1:10,000, Abcam: ab129066; CRMP2, 1:20,000, Abcam: ab129082) and HRP-conjugated goat anti-rabbit (1:15,000, Proteintech Group: SA00001-2, Rosemont, IL, USA) or HRP-conjugated goat anti-mouse secondary antibody (1:15,000, Proteintech Group: SA00001-1, Rosemont, IL, USA). Finally, the bands were measured using Image J software (National institutes of health, Bethesda, USA).

### 4.7. Enzyme-Linked Immunosorbent Assay (ELISA)

Mice were deeply anesthetized with isoflurane and decapitated, and the prefrontal cortex samples were rapidly removed. The prefrontal cortex samples were snap-frozen on dry ice and stored at −80 °C. The prefrontal cortex samples were homogenized in cold RIPA lysis buffer (100 mg/1 mL; Thermo Scientific Pierce, Waltham, MA, USA) with protease inhibitor cocktail (Roche, Indianapolis, IN, USA) and centrifuged at 13,000× *g* for 30 min at 4 °C. The supernatant was collected to analyze the concentration of IL-1β (Abcam, ab100705), IL-6 (Abcam, ab100713), and TNFα (Elabscience, E-EL-M0049c) according to the manufacturer’s instructions. Briefly, 100 µL samples were added into the microplate and incubated overnight at 4 °C. The next day, microplates were washed with wash solution (300 μL) and incubated with the biotinylated IL-1β (IL-6 or TNFα) detection antibody for 60 min. After washing, microplates were incubated with HRP-streptavidin solution for 45 min and incubated with TMB one-step substrate reagent for 30 min. Add 50 μL of stop solution to each well. Absorbance was read at 450 nm using a microplate reader. Values were normalized to volume of supernatant with prefrontal cortex from each individual sample (100 mg/mL).

### 4.8. Slice Electrophysiology

Mice were anesthetized by isoflurane inhalation and decapitated. The brain was removed, trimmed, and embedded in low-gelling-point agarose, and coronal slices (200 μm thick) containing the prefrontal cortex were cut using a vibrating slicer (Leica VT1200s, Nussloch, Germany), as described in our recent studies [62,63]. Slices were prepared in a cutting solution containing the following (in mM): 110 choline chloride, 2.5 KCl, 1.25 NaH_2_PO_4_, 0.5 CaCl_2_, 7 MgSO_4_, 26 NaHCO_3_, 25 glucose, 11.6 sodium ascorbate, and 3.1 sodium pyruvate. After slice cutting, ACSF was progressively spiked into the choline solution every 5 min for 20 min at room temperature to gradually reintroduce Na^+^, similar to a previous method [64]. The slices were allowed to recover for at least an additional 30 min in ACSF prior to recording. All solutions were saturated with 95% O_2_ and 5% CO_2_.

Whole-cell recordings were performed from prefrontal cortex pyramidal neuron using patch-clamp amplifiers (Multiclamp 700B) under infrared differential interference contrast (DIC) microscopy [65,66]. Data acquisition and analysis were performed using DigiData 1440A and 1550B digitizers and the analysis software pClamp 10.7 (Molecular Devices). Signals were filtered at 2 kHz and sampled at 10 kHz. Miniature inhibitory postsynaptic currents (mIPSCs) and miniature excitatory synaptic currents (mEPSCs) were measured based on published studies [67,68,69]. For recording mIPSCs, glutamate receptor antagonists 6-cyano-7-nitroquinoxaline-2,3-dione (CNQX, 20 µM), D-AP5 (50 μM) and TTX (0.5 µM) were present in the ACSF throughout the experiments and pyramidal neurons were voltage-clamped at −70 mV unless otherwise specified. Glass pipettes (3–5 MΩ) were filled with an internal solution containing (in mM): 90 K-gluconate, 50 KCl, 10 HEPES, 0.2 EGTA, 2 MgCl_2_, 4 Mg-ATP, 0.3 Na_2_GTP, and 10 Na_2_-phosphocreatine, pH 7.2 with KOH. For recording mEPSCs, picrotoxin (50 µM) and TTX (0.5 µM) were present in the ACSF throughout the experiments and pyramidal neurons were voltage-clamped at −70 mV unless otherwise specified. Glass pipettes (3–5 MΩ) were filled with an internal solution containing (in mM): 125 cesium gluconate, 10 CsCl, 10 HEPES, 10 EGTA, 1.2 MgCl_2_, 2 Mg-ATP, 0.3 Na_2_GTP and 10 Na_2_-phosphocreatine (pH 7.25).

### 4.9. Statistics

Data analysis was blind to the genotypes and treatment history of the mice. Data are presented as the mean ± SEM. Data sets were compared with two-way ANOVA or Student’s *t*-test followed by Tukey’s post hoc analysis. Post-hoc analyses were performed only when ANOVA yielded a significant main effect or a significant interaction between the two factors. Results were considered to be significant at *p* < 0.05.

## Figures and Tables

**Figure 1 ijms-23-12655-f001:**
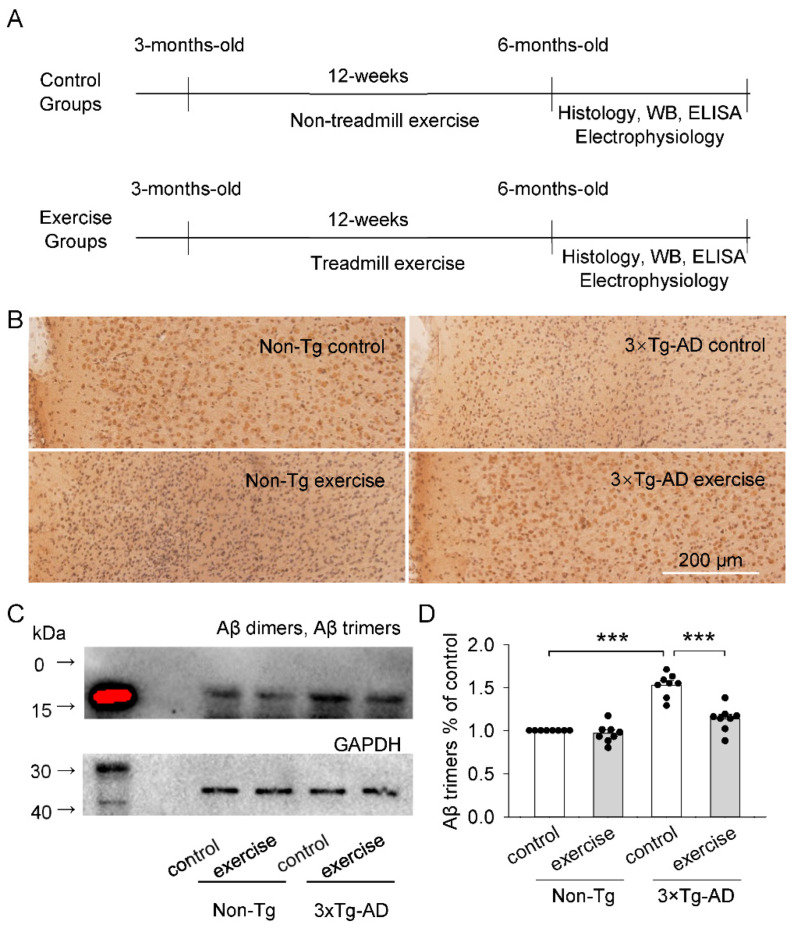
Treadmill exercise alleviated the Aβ pathology of the prefrontal cortex in 3 × Tg-AD mice. (**A**) Timeline of treadmill exercise or non-exercise control, histological test, and electrophysiology. (**B**) Representative microphotographs of the prefrontal cortex sections from 6-month-old mice stained with an anti-Aβ specific antibody (6E10, *n* = 8/group). No Aβ plaques were found in the prefrontal cortex sections. (**C**) Representative western blots for Aβ dimers, Aβ trimers and GAPDH of prefrontal cortex homogenates were prepared from these four groups of mice. Red in (**C**) represents western Blot image overexposure. The Aβ protein content was low, so the marker protein image showed overexposure. (**D**) Summarized data showed that Aβ trimers (*** *p* < 0.001, *n* = 8) in the prefrontal cortex were significantly increased in the 3 × Tg-AD mice compared to the non-Tg control group, and this increase was blocked by treadmill exercise pretreatment (*** *p* < 0.001, *n* = 8). See Appendix A for uncropped blots.

**Figure 2 ijms-23-12655-f002:**
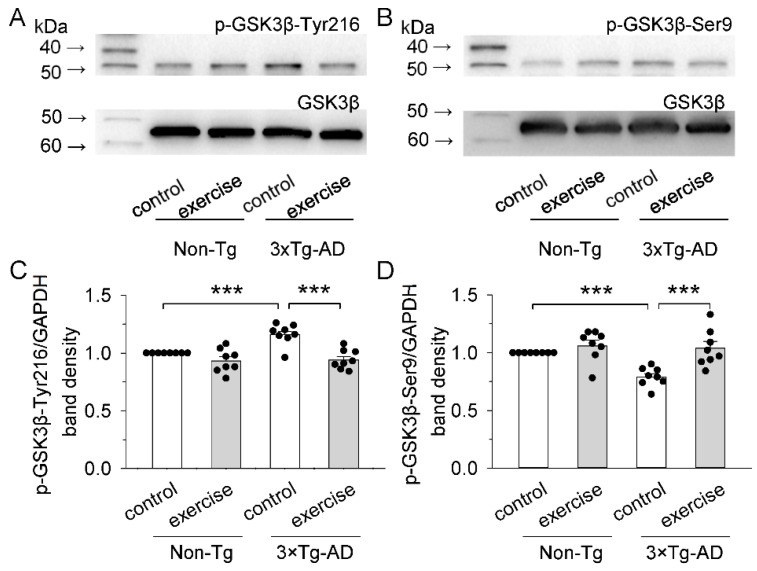
Treadmill exercise inhibited the kinase activity of GSK3β in 3 × Tg-AD mice. (**A**,**B**) Representative western blots for p-GSK3β-Tyr216 (**A**), p-GSK3β-Ser9 (**B**), and GSK3β (**A**,**B**) of prefrontal cortex homogenates were prepared from these four groups of mice. (**C**) Summarized data showed that p-GSK3β-Tyr216/GSK3β in the prefrontal cortex were significantly increased in the 3 × Tg-AD mice compared to the non-Tg control group (*** *p* < 0.001, *n* = 8-8) and this increase was blocked by treadmill exercise pretreatment (*** *p* < 0.001, *n* = 8-8). (**D**) Summarized data showed that p-GSK3β-Ser9/GSK3β in the prefrontal cortex were significantly decreased in the 3 × Tg-AD mice compared to the non-Tg control group (*** *p* < 0.001, *n* = 8-8), and this decrease was blocked by treadmill exercise pretreatment (*** *p* < 0.001, *n* = 8-8). See Appendix A for uncropped blots.

**Figure 3 ijms-23-12655-f003:**
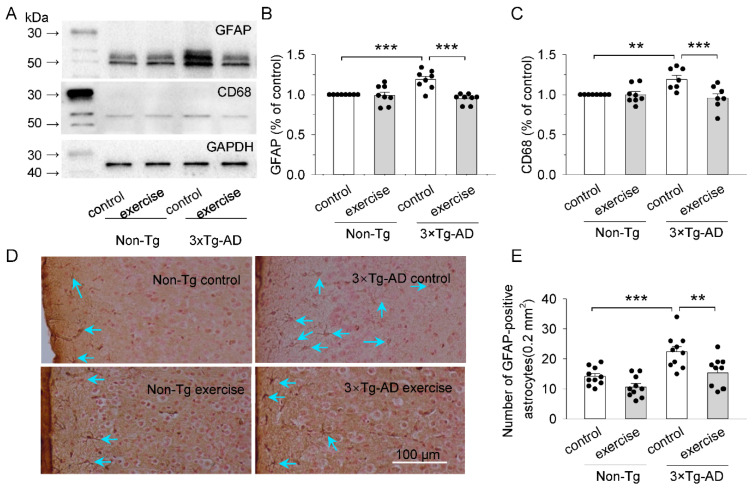
Treadmill exercise reduced astrocyte and microglia activation in 3 × Tg-AD mice. (**A**) Representative western blots for GFAP and CD68 of the prefrontal cortex homogenates were prepared from these four groups of mice. (**B**,**C**) Summarized data showed that GFAP ((**B**), *** *p* < 0.001, *n* = 8-8 and CD68 ((**C**), ** *p* = 0.003, *n* = 8-8) of the prefrontal cortex were significantly increased in the 3 × Tg-AD mice compared to the non-Tg control group and these increases were blocked by treadmill exercise pretreatment (*** *p* < 0.001, *n* = 8-8). (**D**) Representative microphotographs of the prefrontal cortex sections from six-month-old mice. Sections were stained antibody against GFAP (*n* = 10/group). Blue arrows represent GFAP positive cells. (**E**) Quantitative analysis revealed that the number of GFAP-positive cells was significantly higher in 3 × Tg-AD mice compared to the non-Tg control group (*** *p* < 0.001, *n* = 10-10), and this increase was blocked by treadmill exercise pretreatment (** *p* = 0.001, *n* = 10-10). See Appendix A for uncropped blots.

**Figure 4 ijms-23-12655-f004:**
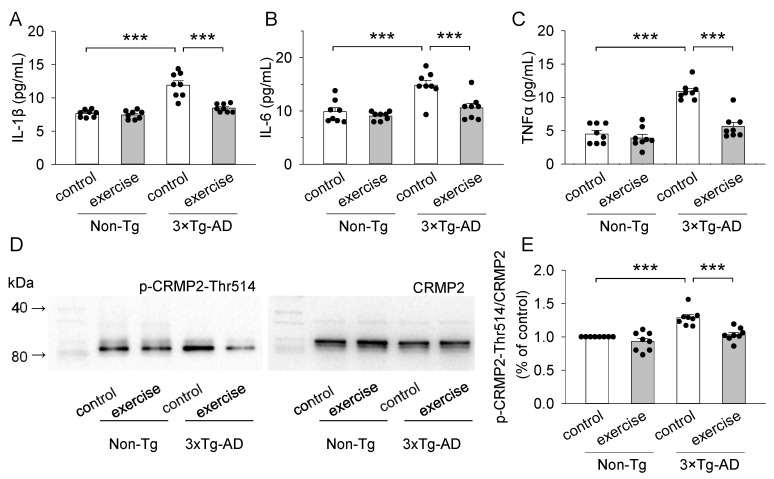
Treadmill exercise decreased the pro-inflammatory cytokines and p-CRMP2 in 3 × Tg-AD mice. (**A**–**C**) ELISA analysis of IL-1β (**A**), IL-6 (**B**), and TNFα (**C**) concentrations in the prefrontal cortex. The concentrations of IL-1β ((**A**), *** *p* < 0.001; *n* = 8-8), IL-6 ((**B**), *** *p* < 0.001; *n* = 8-8), and TNFα ((**C**), *** *p* < 0.001; *n* = 8-8) in the prefrontal cortex were significantly increased in the 3 × Tg-AD control group compared to the non-Tg control group, and these increases were blocked by treadmill exercise pretreatment (*** *p* < 0.001, *n* = 8-8). (**D**) Representative western blots for p-CRMP2-Thr514 and CRMP2 in the prefrontal cortex homogenates were prepared from these four groups of mice. (**E**) Quantitative analysis revealed that the levels of p-CRMP2-Thr514/CRMP2 was significantly higher in 3 × Tg-AD mice compared to the non-Tg control group (*** *p* < 0.001, *n* = 8-8), and this increase was blocked by treadmill exercise pretreatment (*** *p* < 0.001, *n* = 8-8). See Appendix A for uncropped blots.

**Figure 5 ijms-23-12655-f005:**
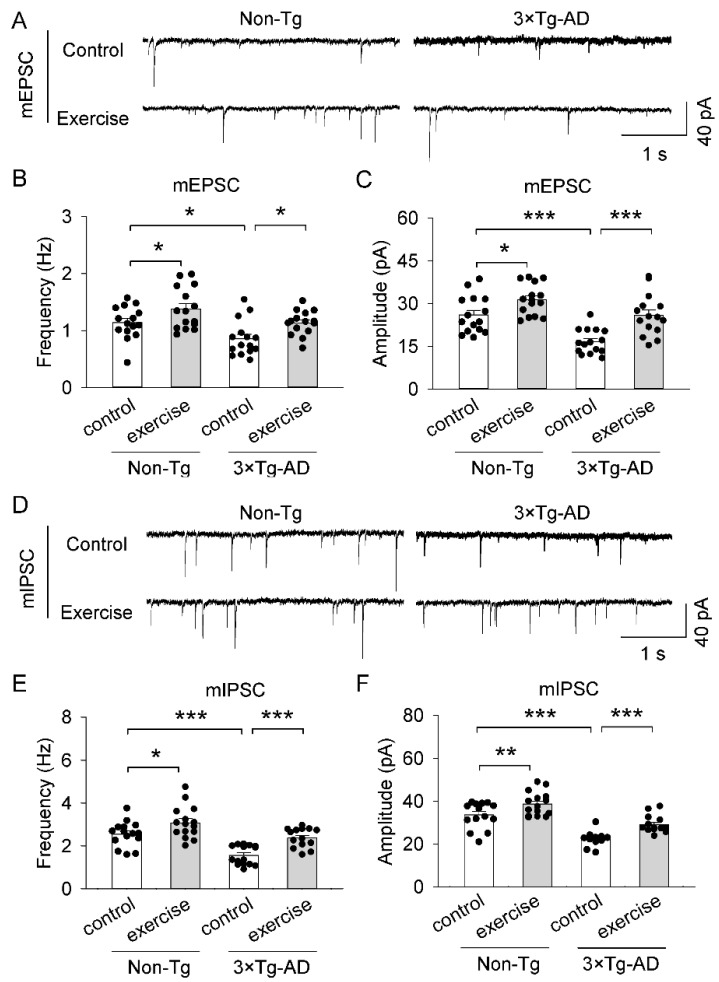
Treadmill exercise blocked the reduction of synaptic transmission of 3 × Tg-AD Mice. (**A**,**D**) Representative mEPSCs (**A**) and mIPSCs (**D**) were recorded from prefrontal cortex pyramidal neurons in slices prepared from the non-Tg control group, non-Tg exercise group, 3 × Tg-AD control group, 3 × Tg-AD exercise group mice. (**B**,**C**) The mean frequency (**B**) and amplitude (**C**) of mEPSCs in the prefrontal cortex in these four groups of mice. The mean frequency (* *p* = 0.014, *n* = 15-15 and amplitude (*** *p* < 0.001, *n* = 15-15) of mEPSCs were significantly decreased in the 3 × Tg-AD mice compared to the non-Tg control group, and these decreases were attenuated by treadmill exercise pretreatment (frequency, * *p* = 0.012, *n* = 15-15; amplitude, *** *p* < 0.001, *n* = 15-15). (**E**,**F**) The mean frequency (**E**) and amplitude (**F**) of mIPSCs in the prefrontal cortex in these four groups of mice. The mean frequency and amplitude of mIPSCs were significantly decreased in the 3 × Tg-AD mice compared to the non-Tg control group (*** *p* < 0.001, *n* = 15-15), and these decreases were attenuated by treadmill exercise pretreatment (*** *p* < 0.001, *n* = 15-15). Treadmill exercise increased the mean frequency (* *p* = 0.012, *n* = 15-15) and amplitude of mIPSC (** *p* = 0.007, *n* = 15-15) in non-Tg mice. Each data set was obtained from 6 mice.

**Figure 6 ijms-23-12655-f006:**
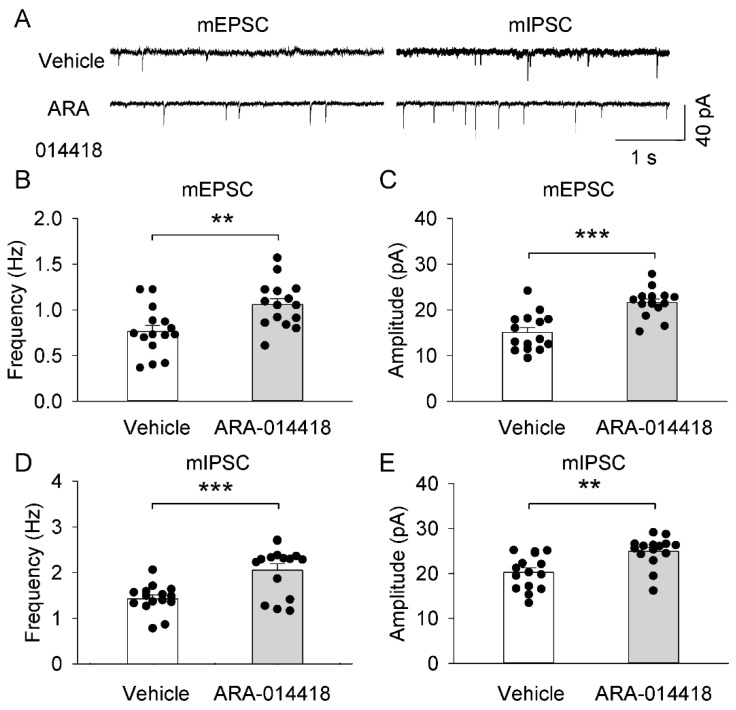
GSK3β inhibitor (ARA-014418) increased synaptic transmissions of 3 × Tg-AD mice. (**A**) Representative mEPSCs and mIPSCs were recorded from prefrontal cortex pyramidal neurons in slices prepared from mice microinjected with vehicle or ARA-014418. (**B**,**C**) The mean frequency (**B**) and amplitude (**C**) of mEPSCs in the prefrontal cortex in these two groups of mice. The mean frequency (** *p* = 0.004, *n* = 15-15) and amplitude (*** *p* < 0.001, *n* = 15-15) of mEPSCs were significantly increased in the ARA-014418 group compared to the vehicle group. (**D**,**E**) The mean frequency (**D**) and amplitude (**E**) of mIPSCs in the prefrontal cortex in these two groups of mice. The mean frequency (*** *p* < 0.001, *n* = 15-15) and amplitude (** *p* = 0.001, *n* = 15-15) of mIPSCs were significantly increased in the ARA-014418 group compared to the vehicle group. Each data set was obtained from 6 mice.

## Data Availability

The datasets used in the analyses described in this study are available from the corresponding author on reasonable request.

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
