# Peer review of "Treadmill Exercise Reduces Neuroinflammation, Glial Cell Activation and Improves Synaptic Transmission in the Prefrontal Cortex in 3 × Tg-AD Mice"

_ijms, 2022, doi:10.3390/ijms232012655_

Round 1
Reviewer 1 Report
In this manuscript, Mu et al., report that treadmill exercise decreases neuroinflammation and improves synaptic transmission in 3xTG-AD mice. The authors found that beta-amyloid, proinflammatory cytokines (IL-1β, IL-6, and TNFa), phosphorylated GSK3 β and collapsin response mediator protein 2 (CRMP2), as well as reactivity of astrocytes are significantly reduced in 3xTG-AD mice that received treadmill exercise for 12 weeks. They also observed that treadmill exercise improves synaptic transmission at both GABAergic and glutamatergic synapses in the prefrontal cortex, which are apparently mediated via GSK3β signaling as ARA-014418, an inhibitor for GSK3β, enhances inhibitory and excitatory synaptic transmission. The findings observed from the present study suggest that exercise might alleviate AD neuropathology. The experiments are well performed and the data are of interest. I only have a couple of minor issues that need to be addressed in the present version of the manuscript.
11. No information is provided in the Materials and Methods section as to whether male or female mice or both were used in the study, as sex is an important biological variable. This information should be included in the Methods section.
2. Authors stated in the manuscript that “treadmill exercise reduces neuroinflammation, glial cell activation and improves synaptic transmission in the prefrontal cortex in 3 × Tg-AD mice, likely via inhibition of GSK3β kinase activity”. While treadmill exercise did reduce phosphorylated GSK3β in 3xTG-AD mice as shown in Figure 2, the authors only performed the experiments where they observed that inhibition of GSK3β with an inhibitor elevates both the frequency and amplitude of mEPSCs and mIPSCs in slice patch-clamp recordings (Figure 6). Therefore, I suggest modifying the conclusion statement.
Author Response
We thank the reviewer for the time and effort in reviewing the manuscript and the insightful comments. Our point-to-point responses are listed below. The revisions in the manuscript are shown by track changes.
- No information is provided in the Materials and Methods section as to whether male or female mice or both were used in the study, as sex is an important biological variable. This information should be included in the Methods section.
Response: We apologize for missing critical details in the Methods and have now clarified that the mice used in this study were males (Line 379-380). Female mice had been used by my other lab members for another study.
- Authors stated in the manuscript that “treadmill exercise reduces neuroinflammation, glial cell activation and improves synaptic transmission in the prefrontal cortex in 3 × Tg-AD mice, likely via inhibition of GSK3β kinase activity”. While treadmill exercise did reduce phosphorylated GSK3β in 3xTG-AD mice as shown in Figure 2, the authors only performed the experiments where they observed that inhibition of GSK3β with an inhibitor elevates both the frequency and amplitude of mEPSCs and mIPSCs in slice patch-clamp recordings (Figure 6). Therefore, I suggest modifying the conclusion statement.
Response: We appreciate the reviewer’s advice. We have revised the conclusion to “treadmill exercise reduces neuroinflammation, glial cell activation and improves synaptic transmission in the prefrontal cortex in 3 × Tg-AD mice” in accordance with the reviewer's comments. (Line 31-32)
Reviewer 2 Report
Reviewer comments and suggestions
The authors in this study investigated treadmill exercise may prevent the pathogenesis and exert neuroprotective effects in 3 × Tg-AD Mice. For this purpose, the authors used techniques such as immunohistochemistry, western blotting, enzyme-linked immunosorbent assay, and slice electrophysiology to analyze various parameters. The study showed that 12-week treadmill exercise beginning in three-month-old mice led to inhibition of GSK3β kinase activity, decreases in the levels of Aβ oligomers, pro-inflammatory cytokines (IL-1β, IL-6, and TNFα), and the phosphorylation of CRMP2 at Thr514, reduction of microglial and astrocyte activation, improvement of excitatory and inhibitory synaptic transmission of pyramidal neurons in the prefrontal cortex of 3 × Tg-AD Mice.
The paper was nicely written, and a few minor comments needed to consider before publication.
- Line 37-43 Please avoid long sentences
- Line 299-300 Did the authors checked (please mention figure or table here)
- Line 323-324 Please mention the table or figure here as well
- Line 358 It would be nice if the authors discuss these two points (GABAergic inhibition and glutamatergic excitation) in the introduction
- Line 362-363 Please check some part was missing in the line
- All references (journal styles should be abbreviated) should be modified based on the MDPI journals. Need to modify all
Author Response
We thank the reviewer for the time and effort in reviewing the manuscript and the insightful comments. Our point-to-point responses are listed below. The revisions in the manuscript are shown by track changes.
- Line 37-43 Please avoid long sentences
Response: We appreciate the reviewer’s advice. We have revised this sentence. (Line 40-46)
- Line 299-300 Did the authors checked (please mention figure or table here)
Response: We have added the relevant figures. (Line 301-302)
- Line 323-324 Please mention the table or figure here as well
Response: We have added the relevant figures. (Line 334)
- Line 358 It would be nice if the authors discuss these two points (GABAergic inhibition and glutamatergic excitation) in the introduction
Response: We have added a brief discussion in the introduction. (Line 37-40)
- Line 362-363 Please check some part was missing in the line
Response: We have added the relevant figures. (Line 374)
- All references (journal styles should be abbreviated) should be modified based on the MDPI journals. Need to modify all
Response: We have revised the format of references based on the MDPI journals.
Reviewer 3 Report
The article titled: „Treadmill exercise reduces neuroinflammation, glial cell activation and improves synaptic transmission in the prefrontal cortex in 3 × Tg-AD mice” by Mu et al. presents studies on 3 × Tg-AD mice and the effects of exercise on parameters involved in AD progression, namely GSK3β, Aβ oligomers (Aβ dimers and trimmers), pro-inflammatory cytokines (IL-1β, IL-6 and TNFα), phosphorylation of CRMP2 at Thr514, and examines synaptic currents in pyramidal neurons in the prefrontal cortex. The research undertaken by the authors addresses the very important context of the contribution of physical exercise to the mechanisms of prevention of the progression of neurodegenerative changes in AD, and thus the possibility of improving the health of patients diagnosed with the disease. The paper is well written, the experiments are correctly planned, I have only a few minor comments, hence I recommend minor correction.
1. Abstract Line 17 “normal” aging - wouldn't “physiological aging” sound better?
2. Please post a figure showing a diagram of the experiments with the age of the mice marked. It took me some time to read at what age the mice studied in the publication were. This is especially difficult because in the journal the materials and methods are at the end of the article, so the figure will be very helpful.
3. Is there a possibility that Aβ dimers were not altered due to the age of the mice? Please address this context in the discussion section. Do the authors plan to study also older mice with developed dementia and significant morphological changes?
4. Fig. 1B change western blot (Aβ dimers, Aβ trimmers) to better quality.
5. Materials and Methods. 4.2. Chemical reagents – please post whether the compounds were used as a solution, if so please explain how they were dissolved or that they were purchased as solutions.
6. 4.6. Western blot. Catalog numbers are missing.
Author Response
We thank the reviewer for the time and effort in reviewing the manuscript and the insightful comments. Our point-to-point responses are listed below. The revisions in the manuscript are shown by track changes.
- Abstract Line 17 “normal” aging - wouldn't “physiological aging” sound better?
Response: We appreciate the reviewer’s advice. We have followed the reviewer’s advice and made corrections. (Line 17)
- Please post a figure showing a diagram of the experiments with the age of the mice marked. It took me some time to read at what age the mice studied in the publication were. This is especially difficult because in the journal the materials and methods are at the end of the article, so the figure will be very helpful.
Response: We appreciate the reviewer’s advice. We have added the timeline in the Figure 1A.
- Is there a possibility that Aβ dimers were not altered due to the age of the mice? Please address this context in the discussion section. Do the authors plan to study also older mice with developed dementia and significant morphological changes?
Response: We agree with the reviewer and appreciate the reviewer’s advice. We have added a brief discussion on the effect of age on Aβ dimers (Line 321-324). Previous studies have shown that Aβ dimers can be found in soluble protein extracts in Tg2576 and J20 AD mice aged 10-14 months. Older mice with developed dementia and significant morphological changes had been studied by my other lab member.
- 1B change western blot (Aβ dimers, Aβ trimmers) to better quality.
Response: We appreciate the reviewer’s advice. We replaced Fig. 1B with better quality image.
- Materials and Methods. 4.2. Chemical reagents – please post whether the compounds were used as a solution, if so, please explain how they were dissolved or that they were purchased as solutions.
Response: We appreciate the reviewer’s advice. The chemical reagents were used as solution. Some of them were prepared as stock solution before use. We have added the following sentence “Picrotoxin was ultrasonic to dissolve in ACSF before use. TTX, CNQX and D-AP5 were dissolved in ddH2O as stock solutions and stored at -20°C freezer. ARA-014418 was dissolved in DMSO at stock concentrations of 50 mM and stored at -80°C freezer.” in the Materials and Methods section (Lines 392-395).
- 4.6. Western blot. Catalog numbers are missing.
Response: The catalog numbers of all antibodies used in western blot were added (Lines 447-454).